# *Staphylococcus aureus* Host Tropism and Its Implications for Murine Infection Models

**DOI:** 10.3390/ijms21197061

**Published:** 2020-09-25

**Authors:** Daniel M. Mrochen, Liliane M. Fernandes de Oliveira, Dina Raafat, Silva Holtfreter

**Affiliations:** 1Department of Immunology, Institute of Immunology and Transfusion Medicine, University Medicine Greifswald, Ferdinand-Sauerbruch-Strasse DZ 7, 17475 Greifswald, Germany; liliane.fernandes@med.uni-greifswald.de (L.M.F.d.O.); dina.raafat@med.uni-greifswald.de (D.R.); silva.holtfreter@med.uni-greifswald.de (S.H.); 2Department of Microbiology and Immunology, Faculty of Pharmacy, Alexandria University, 21521 Alexandria, Egypt

**Keywords:** *Staphylococcus aureus*, host adaptation, mouse models, humanized mice, microbiome, mouse-adapted, JSNZ, wildling, dirty mouse, vaccine

## Abstract

*Staphylococcus aureus* (*S. aureus*) is a pathobiont of humans as well as a multitude of animal species. The high prevalence of multi-resistant and more virulent strains of *S. aureus* necessitates the development of new prevention and treatment strategies for *S. aureus* infection. Major advances towards understanding the pathogenesis of *S. aureus* diseases have been made using conventional mouse models, i.e., by infecting naïve laboratory mice with human-adapted *S.*
*aureus* strains. However, the failure to transfer certain results obtained in these murine systems to humans highlights the limitations of such models. Indeed, numerous *S. aureus* vaccine candidates showed promising results in conventional mouse models but failed to offer protection in human clinical trials. These limitations arise not only from the widely discussed physiological differences between mice and humans, but also from the lack of attention that is paid to the specific interactions of *S. aureus* with its respective host. For instance, animal-derived *S. aureus* lineages show a high degree of host tropism and carry a repertoire of host-specific virulence and immune evasion factors. Mouse-adapted *S.*
*aureus* strains, humanized mice, and microbiome-optimized mice are promising approaches to overcome these limitations and could improve transferability of animal experiments to human trials in the future.

## 1. Staphylococcus aureus

*Staphylococcus aureus* (*S. aureus*) was first described in 1880 by the Scottish surgeon Alexander Ogston, who isolated this bacterium from a wound and demonstrated its importance as a pus pathogen [1]. *S. aureus* is a coccal (Greek kókkos, “the grain”), Gram-positive bacterium of about 1 μm diameter, which arranges itself in dense clusters (Greek staphylé, “the grape”). Colonies on blood agar typically develop an ivory to golden yellow (Latin aureus, “golden”) pigmentation due to the carotenoid staphyloxanthin [2]. *S. aureus* is capable of expressing coagulase, which occurs in multiple allelic variations within the species; a property that is used to distinguish it from its less pathogenic “next of kin”, the coagulase-negative staphylococci [3,4].

About 20–30% of individuals are persistently and asymptomatically colonized with *S. aureus* in their nose, while the remainder constitutes intermittent carriers, i.e., phases of colonization alternate with phases of non-colonization [5,6,7]. Upon invasion, *S. aureus* can cause a variety of diseases, ranging from minor skin and soft tissue infections such as impetigo, folliculitis, and cutaneous abscesses, to life-threatening diseases such as sepsis, infective endocarditis, or toxic shock syndrome [8]. The emergence and continuing spread of multi-resistant *S. aureus* strains, such as methicillin-resistant *S. aureus* (MRSA) and vancomycin-resistant *S. aureus* (VRSA), complicate the treatment of staphylococcal infections and cause a significant economic burden. For example, a multicenter study in China showed an average increase in hospital costs of 3182 USD to 6533 USD for MRSA patients as compared to methicillin-sensitive *S. aureus* (MSSA) patients [9]. Coupled to the fact that even reserve antibiotics fail to successfully treat severe *S. aureus* infections, it is not surprising that the WHO lists *S. aureus* as one of the major health threats in a so-called “post-antibiotic era” [10]. Hence, new antibiotics and alternative prophylactic or therapeutic strategies are urgently needed to combat *S. aureus* and other so-called ESKAPE bacteria, i.e., nosocomial pathogens that show multidrug resistance and a high virulence [11,12,13].

Nosocomial infections with MRSA, also called hospital-associated MRSA (HA-MRSA), have been known for a long time, and thus, effective measures have been taken to limit their spread [14,15,16]. Since HA-MRSA rarely cause disease in healthy individuals without predisposing conditions, it has been suggested that they display a reduced virulence and could therefore only thrive in low-competition environments [17]. Hence, they pose a serious risk to immunocompromised people, rather than the general population. However, the emergence of highly virulent community-associated MRSA (CA-MRSA) and livestock-associated MRSA (LA-MRSA) revealed that even immunocompetent individuals are at risk of severe MRSA infection [15]. The first described CA-MRSA cases date back to the 1980s, and since then, regionally dominant strains have evolved, with USA300 being the prime culprit in Northern America. In contrast to HA-MRSA strains, most CA-MRSA strains possess other staphylococcal cassette chromosome mec (SCCmec) variants, as well as fewer and varying resistance determinants, and are more likely to carry the Panton-Valentine leukocidin, a two-component toxin directed against human neutrophils [14,15,16,18]. LA-MRSA were first described in 2005 by Voss et al., who demonstrated their transmission between pigs and pig farmers [19]. Meanwhile, numerous LA-MRSA strains were discovered in different animal hosts, and their virulence potential seems to be comparable to human strains [20,21]. It is therefore safe to assume that *S. aureus* is not only dependent on human-to-human transmission but also holds zoonotic potential.

The emergence of CA-, LA-MRSA, and the increasing number of *S. aureus* findings in wild animals has important implications for dealing with *S. aureus* [22,23]. Since serious *S. aureus*/MRSA infections are not only limited to the hospital setting and immunocompromised patients but also affect healthy individuals in the general population, and since *S. aureus* is not only a human commensal and pathogen but also a zoonotic, widespread agent with many animal reservoirs in husbandry and the environment, a more comprehensive approach is needed to disrupt *S. aureus* infection chains and to find proper treatment strategies. The “One Health” concept with its holistic approach of integrating human, animal, and environmental health, therefore, seems ideal for dealing with the multi-faceted lifestyle of *S. aureus* [24].

The aim of this review is to provide an overview of the adaptive and dynamic nature of *S. aureus* in terms of its spread and host adaptation. We want to highlight the intrinsic and ecological features that allow the spread of *S. aureus* among a multitude of hosts, and to discuss whether this makes *S. aureus* a generalist or a specialist when it comes to host adaptation. The adaptive features that arise from this lifestyle of *S. aureus* and its specific interaction with the respective host have serious implications for *S. aureus* research, especially regarding mouse models of infection. We will outline the shortcomings of conventional mouse models, i.e., those using established laboratory mice such as C57BL/6 or BALB/c in combination with human-adapted *S. aureus* strains such as Newman. The use of such models has often hampered the transferability of animal testing to the human system and thereby delayed much-needed progress in *S. aureus* research. Finally, we will discuss alternative approaches for murine models that take the strong interspecies interaction of *S. aureus* and its host into account.

## 2. *S. aureus*’ Extended Host Spectrum

*S. aureus* is a notorious multi-host pathogen. Apart from humans, it has been successfully isolated all over the world from a variety of companion, farm, and wild animals, which has recently been reviewed thoroughly [22,25,26,27]. Even laboratory animals, such as mice and rats, are sometimes “naturally” colonized or infected with *S. aureus* in their breeding facilities [28,29,30]. Nevertheless, by comparing the number and complexity of *S. aureus* lineages in humans and other hosts, and by reconstructing the emergence of *S. aureus* in those hosts, it becomes evident that humans are the primary host of *S. aureus*, acting as a major hub for numerous host switches [31]. 

Judging by this vast host spectrum of *S. aureus*, one is tempted to consider *S. aureus* as a pure generalist [32]. This, however, neglects the population structure of *S. aureus*, which might shed more light on *S. aureus* host tropism. *S. aureus* has a clonal population structure, which is characterized by genetic clonality. Genetic clonality itself is a result of clonal evolution, whose defining property is a relatively low level of recombination, leading to two phenomena: (i) linkage disequilibrium, i.e., the nonrandom association of alleles at different loci, as opposed to the independent, random distribution of these alleles according to their frequencies in the population; and (ii) a strong phylogenetic signal, i.e., the tendency of related lineages to resemble each other more than lineages drawn randomly from the same tree (near-clading) [31,33,34,35,36]. This results in a cluster-like arrangement of family trees showing different *S. aureus* strains. When using multi-locus sequence typing (MLST) as the typing method, these clusters are formed by different *S. aureus* sequence types (STs) and are referred to as clonal complexes (CCs) [36,37]. CCs usually exhibit a high genomic similarity and share common traits that can distinguish them from other CCs.

As pertains to the species specificity, there are major differences between CCs. Recent molecular epidemiological studies set out to define a host range for each *S. aureus* CC. While some CCs are able to thrive in a broad range of hosts, others show a very restricted host spectrum. For example, CC398 strains, which were generally considered to be pig-associated, have also been isolated from humans, horses, poultry, sheep, rabbits, and other animals. Such strains are occasionally referred to as extended-host-spectrum genotypes (EHSG), emphasizing their apparent lack of specific host tropism and their ability to colonize/infect several species [38]. In contrast, lineages such as CC121, CC385, and CC522 seem to possess a much narrower host spectrum and are found in either one or very few species [23,25]. Such occasional findings of *S. aureus* in few animals might reflect either (i) the presence of highly host-adapted, specialist *S. aureus* strains; (ii) a “sampling bias”, where additional reservoirs still remain to be investigated; or (iii) spillover events.

However, host adaptation cannot be easily distinguished from spillover events without further investigation. Following a concept from ecology that describes the interaction of hosts with their parasites (and pathogenic bacteria can surely be considered as such), two barriers must be overcome for a successful and lasting host change: the encounter and the compatibility filter. Encounter filters, such as living in distinct geographic regions or behavior that excludes interaction, prevent contact between host and bacterium, while compatibility filters are barriers that originate from physiological and biochemical incompatibilities as well as the host’s immune system once contact has been made [39,40]. If one considers the ubiquitous distribution of *S. aureus*, the colonization/infection of numerous species, many of which can serve as vectors as well, and the manifold ways of transmission, may it be human–human, human–animal, animal–animal, insect-borne, via animal products, or environmental routes such as water, soil, air, or manure, it becomes clear that the encounter filter has basically collapsed and only the compatibility filter remains [22,26,41,42]. Moreover, most ecosystems have a high permeability for animals such as mice or birds that also serve as bridge hosts, connecting species living in distinct ecosystems, thereby enhancing the spread of *S. aureus* [43]. This is an ideal situation for spillover events, when *S. aureus* jumps from one host to another and transiently colonizes/infects it, without adapting to it and therefore fails to establish a sustained transmission within the new host species. Thus, there is no change of host tropism, and once the source of transmission disappears, *S. aureus* will most likely die out in that host. Spillover events could therefore occur quite frequently, while an actual host switch is rare [42]. Therefore, the mere occurrence of *S. aureus* in certain hosts does not necessarily imply host adaptation. The latter requires thorough genetic and phenotypic studies to affirm its presence.

## 3. Host Adaptation of *S. aureus*

Adaptation to a new host is usually a multifactorial process that requires specific modifications of the bacterial genome for successful establishment in the new host niche [44]. The genome of *S. aureus* can be divided into three parts (with the percentage of the respective part of the total genome written in brackets):Core genome (75%), which is quite conserved in all lineages and codes for housekeeping genes, genes of central metabolism, and some conserved virulence factors such as α-hemolysin (*hla*);Accessory genome (15%) that consists of mobile genetic elements (MGE), such as transposons, prophages, pathogenicity islands, plasmids, and chromosomal cassettes;Core variable genome (10%), which contains mostly genes encoding surface proteins and some virulence regulators [45,46,47,48,49].

Depending on the relatedness of two host species between which *S. aureus* moves, the host environmental factors can change drastically. Surface receptors for attachment of *S. aureus* [50,51,52], nutrient availability [31], properties of the immune system [53], coagulation factors involved in immune evasion [54,55], and many more factors can differ in a way that requires genetic modification of all three parts of the *S. aureus* genome [27,56]. Hence, adaptation to a new host occurs through adaptive evolution and is marked by genomic decay, genomic rearrangements, and gene acquisition as typical genomic signatures [44]. Concerning the adaptation of *S. aureus* lineages to different animal hosts, many research groups have successfully identified these signatures, for instance in the form of point mutations in all parts of the *S. aureus* genome, acquisition or loss of MGE via horizontal gene transfer or pseudogenes, as has recently been reviewed by Matuszewska et al. and Haag et al. [25,27]. Briefly, typical findings are (i) the loss or inactivation of human-specific virulence factors such as the immune evasion cluster (IEC), which is highly prevalent in human isolates; (ii) the acquisition of virulence genes that act highly host-specific, e.g., a modified *vwb* in bovine isolates that allows the coagulation of ruminant plasma; and (iii) modification of resistance patterns via MGE-acquired genes or mutation with respect to the host environment. Furthermore, point mutations and recombination in genes as well as regulatory elements linked for instance to nutrient supply, transcriptional regulation, and virulence have been reported [31,57,58]. The immune system is probably the most important driver of host adaptation and, thus, a main reason for diversification of *S. aureus* lineages [40]. From the moment it enters an organism, *S. aureus* is exposed to antimicrobial peptides, complement, phagocytes, antibodies, T cells, and other humoral and cellular factors set to eradicate the bacterium. In order to survive, *S. aureus* has developed intricate, redundant, and highly host-specific immune evasion mechanisms that differ among the various CCs [31,59,60]. 

As host-adaptive changes are usually limited to a single or few closely related *S. aureus* lineages, a phylogenetic clustering by the host can be observed, which—together with the aforementioned phenomena—speaks for a specialist-like behavior of certain *S. aureus* lineages. In conclusion, it is safe to assume that *S. aureus* as a species is a generalist bacterium, as reflected by its broad host range and lineages belonging to the EHSG. Nevertheless, some lineages act as specialists, showing strong host tropism and an adapted phenotype [25]. 

## 4. Implications of Staphylococcal Host Adaptation for Murine Colonization and Infection Models

The mouse is the most commonly used animal to model human infections and diseases, for a variety of reasons. Mice are small and cheap, do not require significant space, and are easy to maintain and breed. Furthermore, the availability of a multitude of transgene, knockout, and knockin mice renders them quite lucrative for animal research [61,62,63,64]. 

Since their divergence around 75 million years ago, humans and mice have evolved independently from each other and therefore accumulated significant genomic differences [65]. Nevertheless, they still share similarities regarding their physiology, e.g., within the cardiovascular, endocrine, nervous, and immune systems, which in turn is reflected in the great advances made in human health through the use of mouse models in those fields [61]. Still, whether mice are suitable models for human diseases is heavily debated within the scientific community, since similarities do not necessarily translate into a successful transfer to the human system. This has been evidenced by many studies, which were initially promising in the mouse model but failed when applied to humans [66,67,68,69,70,71,72]. The significant differences in both the adaptive and innate immune systems of mice and humans, discussed in detail elsewhere [53,73,74,75,76], are among the main reasons for such a failure. The following sections will highlight the problems of conventional mouse models involving *S. aureus* and will suggest possible solutions and alternatives.

### 4.1. Current Problems with Conventional S. aureus Mouse Models

Conventional mouse models are the most frequently used systems for investigating a variety of *S. aureus* diseases, e.g., skin and soft tissue infections [77,78], bacteremia [79,80], sepsis [81,82], peritonitis [83,84], pneumonia [85,86], osteomyelitis [87,88], and endocarditis [89,90]. Although these models led to important advancements in our understanding of host–pathogen interaction, as well as to the identification of key virulence factors and potential treatment strategies for *S. aureus* infections, scientists have had to deal with failures of human clinical trials based on mouse models, with vaccination studies being a prime example [91,92,93]. So far, all vaccine candidates have failed in human clinical trials, resulting in a lack of an effective vaccine against invasive *S. aureus* infections, which in turn poses a serious health threat to humanity in view of the spread of multi-resistant *S. aureus* strains [94,95]. The potential reasons for this lack of transferability between murine and human trials are plentiful but not always easy to delineate, hence requiring that each model undergoes its own thorough adaptation.

As outlined below in detail, several factors can impact the outcome of murine *S. aureus* infection models. These include (i) genetic variations among *S. aureus* isolates; (ii) genetic variations among mouse strains; (iii) infectious dose; (iv) murine microbiome; (v) constitution and function of the immune system of mice and humans; (vi) species-specific staphylococcal toxins and immune evasion molecules; as well as (vii) species-specific staphylococcal adhesins and nutrient acquisition systems.

#### 4.1.1. Genetic Variations among *S. aureus* Isolates

A major confounding factor is the choice of the *S. aureus* strain. Indeed, using different *S. aureus* strains has often lead to different results in the same animal model [86,89,96,97,98,99,100,101,102,103,104] (Table 1). Considering that about 25% of the *S. aureus* genome is variable and that genome-wide adaptive point mutations and recombinations occur frequently, it is not surprising that virulence varies significantly between *S. aureus* lineages. Differences in the genomic setting of *S. aureus* strains, resulting in changes in metabolic requirements, protein production including the production of different virulence factors, and transcription regulation have clearly been linked to a modulation of virulence [31,101,105,106,107,108,109]. Hence, the choice of *S. aureus* strains massively influences the experimental outcome and should therefore be carefully considered. Consequently, although a rather costly and labor-intensive approach, it may be appropriate to validate major experimental findings by testing a variety of relevant *S. aureus* strains. Furthermore, genomic and transcriptomic analyses might help delineate the influence of certain virulence factors and transcriptional regulation in defined disease settings.

#### 4.1.2. Genetic Variations among Mouse Strains 

Not only the choice of *S. aureus* strains impairs comparability and transferability but also the choice of the mouse strain. In a model using *S. aureus* SH1000, Köckritz-Blickwede et al. showed that A/J, BALB/c, C57BL/6, C57BL/10, C3H/HeN, CBA, and DBA/2 mice differ in their resistance towards intravenous *S. aureus* infection. C57BL/6 mice had the highest resistance with respect to controlling bacterial growth and survival, followed by C57BL/10, C3H/HeN, and CBA mice, while A/J, BALB/c, and DBA/2 mice were highly susceptible and died shortly after bacterial infection [110]. On the other hand, in a subcutaneous infection model also using *S. aureus* SH1000, Nippe et al. demonstrated an enhanced susceptibility of C57BL/6 mice compared to BALB/c and DBA/2 mice. This was attributed to BALB/c and DBA/2 mice reacting via a Th2 response, while C57BL/6 mice reacted via a Th1 response, rendering them more susceptible [111]. It is well known that certain mouse strains possess such a bias towards a Th1 or Th2 response [112]. Coupled to the fact that the infection site often governs the type of Th response necessary to combat *S. aureus*, and based on previous findings in humans or other animal models, a careful selection of the mouse strain would be advisable [113,114,115], to avoid conflicting or misleading results [111,116].

#### 4.1.3. Infectious Dose

To establish an *S. aureus* infection in mice, a high infectious dose (10^6^–10^9^ CFU) is generally required, which might even vary greatly between different models [93,117,118]. Data pertaining to the minimal infectious doses for human *S. aureus* diseases are scarce. Schmid-Hempel et al. suggested that 10^5^ CFU are necessary for infection but did not further substantiate that assumption with published studies and seem to mainly focus on staphylococcal food poisoning [119,120]. In 1971, Singh et al. published a study in which the skin of human volunteers was exposed to different inocula of *S. aureus*, where as little as 40 CFU/cm^2^ were deemed sufficient to establish an infection in 20% of subjects. By raising the inoculum size to 2000 CFU/cm^2^, the infection rate could be increased up to 67% [121]. Taking into account the “Trojan horse” hypothesis, which relates to the ability of *S. aureus* to hijack circulating immune cells, such as neutrophils, to be able to travel to and infect other body sites, it becomes reasonable to assume that even a small initial number of bacteria might be just enough to establish an active focus of infection [122,123,124,125]. Hence, the evidently high number of bacteria that is used in conventional mouse models might obscure processes that are relevant to those early, low-bacterial burden stages of infection. This is further corroborated by the fact that certain virulence factors of *S. aureus* are regulated by quorum sensing mechanisms and are therefore under the influence of bacterial density [126,127]. Moreover, a higher initial bacterial number is accompanied by the production of large concentrations of bacterial products including, for instance, pathogen-associated molecular patterns, which in turn could start a strong pro-inflammatory immune response and thereby deviate from the expected immune response to a low infectious dose. 

The influence of subtle but measurable changes in infectious dose has been clearly demonstrated in a model of *S. aureus* prosthetic joint infection by Vidlak et al., who compared the immune responses in wildtype (C57BL/6) and IL-12p40 knockout mice after infection with 10^3^ or 10^5^ CFU. Notably, there were significant differences between wildtype and knockout mice in the recruitment of myeloid-derived suppressor cells, monocytes, macrophages and T cells to the infection site, as well as in the bacterial burden of the joint, surrounding tissue and the femur after seven days of infection with 10^3^ CFU. However, those differences were no longer observed with the 10^5^ CFU infectious dose. Similarly, differences in cytokine and chemokine responses were nullified by the higher infectious dose 14 days post infection [117]. Such studies highlight the fact that serious efforts must be undertaken to lower the infectious dose in murine models of *S. aureus* infection. Otherwise, one might run the danger of masking crucial aspects of the immune response, thus deviating it even further away from the human system.

#### 4.1.4. Murine Microbiome

The mouse microbiome is yet another factor affecting the transferability of mouse models. Within recent years, more and more evidence has been collected, showing that the microbiome is an etiologic agent of certain diseases, such as metabolic disorders or inflammatory bowel disease, is implicated in normal human physiology and also interacts strongly with the immune system [62,128,129,130]. From an ecological point of view, the latter aspect is not surprising, since it is beneficial for both host and commensal to co-evolve toward mutualism and homeostasis. A prerequisite for this is a finely tuned balance between the growth of commensal microorganisms and the prevention of their invasion of the body or overuse of resources, which is known as tolerance. The immune system ensures this under homeostatic conditions and is therefore both shaped and modulated by the microbiome throughout life, as comprehensively reviewed by Zheng et al. [130]. As the microbiomes of mice and humans differ significantly, this also leads to immunological differences between the two species that affect the interaction with potential pathogens such as *S. aureus* [62,128]. 

This problem is aggravated in laboratory mice that live under extremely artificial conditions, with active exclusion of certain microorganisms, which would otherwise influence the immune response, as can be observed either in accidental or planned exposure [131]. Even small changes in the murine microbiome can affect resistance to or infectivity of *S. aureus*. In a lung infection model, mice whose intestinal microbiome lacks segmented filamentous bacteria (sfb) develop a more severe *S. aureus* pneumonia characterized by a higher bacterial load, inflammation, and mortality compared to sfb-positive mice. This can be reduced by the acquisition of sfb by sfb-negative mice [132]. Conversely, the co-injection of *Micrococcus luteus*, a typical human skin commensal, with *S. aureus* into mice augments *S. aureus* pathogenicity [133]. 

The strong interaction of *S. aureus* with typical microbiota has also been demonstrated in human studies. Iwase et al. showed that the serine protease Esp, which is produced by the commensal *Staphylococcus epidermidis,* inhibits *S. aureus* biofilm formation and nasal colonization [134]. Other bacteria interfering with *S. aureus* colonization are, for instance, *Staphylococcus lugdunensis* via a peptide antibiotic called lugdunin [135], as well as certain *Bacillus* species that produce fengycin lipopeptides, which interfere with *S. aureus* quorum sensing and thereby disrupt colonization [136]. 

The aforementioned examples clearly show the importance of microbiota in *S. aureus* colonization and infection. Bacterial interference, competition for nutrients and niche competition are just some aspects of bacteria–bacteria interaction that should be considered in mouse models. Humanizing the mouse microbiome or adapting it more to the natural murine microbiome, and thereby its natural environment, might therefore allow for more reliable results, depending on the research question asked.

#### 4.1.5. Constitution and Function of the Murine and Human Immune Systems 

As previously mentioned, the immune systems of mice and humans display distinct differences, which are attributed not only to different host–commensal and host–pathogen interactions but also to anatomical, physiological, biochemical, and ecological differences that imprinted themselves over the course of evolution [62]. For instance, and in contrast to humans, mice have significant bronchus-associated lymphoid tissue, which might reflect an enhanced exposure to environmental antigens, since their small body size places them much closer to the ground [53]. Further differences include, for example, composition and function of the innate and adaptive immune cell repertoire, a differential expression of surface proteins such as Toll-like receptors and MHCII, and the presence of leukocyte defensins, as reviewed in detail elsewhere [53,73,74,75,76]. This has to be considered when transferring murine results to the human system. Neutrophils, one of the first lines of defense against *S. aureus* infections and therefore crucial for disease development, are much less abundant in murine peripheral blood, which might impact the phagocytic capacity of mice compared to humans [53]. Another example are γδ-T cells that strongly contribute to protection in murine *S. aureus* pneumonia by accumulating in the lung and releasing IL-17 upon activation but are much less prevalent in human tissues [137,138].

#### 4.1.6. Species-Specific Staphylococcal Toxins and Immune Evasion Molecules

Another important drawback of conventional mouse models concerns toxins and immune evasion molecules targeting species-specific sequence variants of immune cell receptors or humoral factors of the immune system. There is growing evidence that several virulence factors that are important in human *S. aureus* strains are ineffective or less effective in the mouse. Superantigens are toxins that activate large fractions of T cells by crosslinking T cell receptors with major histocompatibility complexes independent of their antigen specificity and contribute to immune evasion in humans [139]. Due to sequence differences in murine and human MHCII, superantigens are several orders of magnitude less potent in mice [140,141,142]. Similarly, leukocidins of *S. aureus* show a reduced activity in mice [143,144]. For instance, the bicomponent pore-forming leukotoxin LukAB targets human phagocytes but not their murine counterparts, due to host-specific variants of its targeted receptor CD11b [145]. Since phagocytes, especially neutrophils, are of utmost importance in *S. aureus* pathogenesis, judged by the vast immune evasion repertoire of *S. aureus* specifically directed against those cells, non-functional virulence factors could have a huge impact on disease development and progression. Factors impairing neutrophil function that are active in humans but not, or to a lesser degree, in mice are, for instance, γ-hemolysins and CHIPS [146,147]. In addition, the highly human-specific and among human isolates widespread IEC, containing several immune evasion factors that promote complement evasion and establishment of infection, is basically absent in most animal isolates [148,149]. Interestingly, IEC-encoding phages integrate into the β-hemolysin gene (*hlb*) and thereby abrogate β-hemolysin production. Depending on the host species-associated selective pressures, the ability to produce either the IEC-encoded immune evasion factors or β-hemolysin appears to be of greater advantage. The vast majority of human strains carry IEC-encoding prophages, implying that the complement-blocking effects of the IEC factors outweigh the disadvantage of *hlb* inactivation. Animal strains, on the other hand, do not seem to benefit from the IEC to this extent, and thus tend to eliminate IEC-encoding phages and restore *hlb*. Concordantly, Katayama et al. observed the loss of the IEC-encoding prophage in *S. aureus* MW2 during adaptation to murine skin. As a result, *hlb* was restored, and the strain started to produce Hlb, which promoted murine skin colonization by *S. aureus* more than 50-fold [150]. This example illustrates how the interaction of *S. aureus* with the immune system can generate host-specific selective pressures.

On the contrary, animal-specific *S. aureus* isolates also harbor host-specific virulence or immune evasion factors that are not active on human cells. For example, bovine isolates frequently encode the pore-forming toxin LukMF’, which kills bovine but not human neutrophils, because its cellular receptor, CCR1, is present on bovine but absent on human neutrophils [151].

In general, it can be said that the interaction with the host’s immune system shapes the virulence factor repertoire of *S. aureus* and adapts it accordingly. Consequently, different immune systems favor different virulence characteristics. This complicates transferability between human and murine systems, as host–pathogen interaction can vary drastically, depending on the strains used.

#### 4.1.7. Species-Specific Staphylococcal Adhesins and Nutrient Acquisition Systems

Moreover, *S. aureus* shows strong host tropism in the interaction with host molecules that are involved in pattern recognition, adherence, and nutrient acquisition of *S. aureus*. The pattern recognition receptor langerin interacts with *S. aureus* through conserved β-N-acetylglucosamine modifications on wall teichoic acid (WTA). Compared to human langerin, its murine isoform binds WTA 10–100-fold weaker [152]. The same is true for the staphylococcal adhesin Fnbp. Fnbp derived from a human strain shows a weaker binding towards murine than human fibrinogen [61]. Another host-specific factor is the bacterial iron uptake system. Iron acquisition is critical for the survival of *S. aureus* within its host and is partly achieved through iron extraction from hemoglobin. Hemoglobin is bound via staphylococcal IsdB, which binds human hemoglobin significantly stronger (K_D_ = 5.5 × 10^−8^ M) than murine hemoglobin (K_D_ = 9.8 × 10^−7^ M) [153]. As a result, human *S. aureus* isolates would suffer from iron shortage when used in mice.

In summary, it can be said that conventional mouse models exhibit several limitations that hamper transferability of results to the human system. The choice of the *S. aureus* and mouse strains, the infectious dose, differences between mice and humans in microbiomes and immune systems, and differential interactions with host tissues and molecules strongly influence the experimental outcome. Thus, solutions must be found to resolve the incompatibilities between human *S. aureus* strains and laboratory mice, since as Kim et al. put it: “Mice are, however, not a natural host for human clinical *S. aureus* isolates (…)” [93].

### 4.2. Alternative Approaches to Conventional Mouse Models

In an attempt to circumvent apparent incompatibilities between human *S. aureus* strains and laboratory mice, scientists have been on the search for a more suitable animal model. In recent decades, many such *S. aureus* infection models emerged, involving (i) invertebrates such as the fruit fly (*Drosophila melanogaster*), the roundworm (*Caenorhabditis elegans*) and the honeycomb moth (*Galleria mellonella*); (ii) vertebrates, e.g., zebrafish, rats, rabbits, sheep, dogs, goats, pigs, guinea pigs, and hamsters; as well as (iii) nonhuman primates [154,155,156,157,158,159]. However, for a variety of reasons, none of the above has proven to be generally more suitable for *S. aureus* research. 

From an ethical perspective, the use of invertebrates in virulence models would be preferable; moreover, they could help to make follow-up experiments more targeted [156,157]. However, due to the significant anatomical, physiological, and immunological differences between invertebrates and vertebrates, these models will remain unsuitable for studying more complex host–pathogen interactions [160,161]. Other vertebrate models essentially exhibit the same limitations addressed in Section 4.1.1, Section 4.1.2, Section 4.1.3, Section 4.1.4, Section 4.1.5, Section 4.1.6 and Section 4.1.7. Nevertheless, depending on the research question at hand, they might be more suitable than murine models when investigating certain virulence factors that show a low activity in mice, such as the use of rabbits for studying PVL, HlgCB, or modified LukAB [143,144,162]. Because of their close relatedness to humans, nonhuman primates would theoretically be an excellent model for *S. aureus* research [163,164,165], but—apart from serious ethical issues—the immense financial and laborious burdens associated with such experimental settings often deter most laboratories.

For all of the above mentioned reasons, and for lack of better alternatives, we will focus in the following sections on improvements made in mouse models for *S. aureus* research.

#### 4.2.1. Improving *S. aureus* Mouse Models by Using Host-Adapted *S. aureus* Strains

The usage of host-adapted strains might solve some of the problems mentioned in Section 4.1. and is already implemented for several bacteria besides *S. aureus.* For instance, host-adapted *Helicobacter pylori* strains can be generated by serial passage of clinical isolates in mice and induce gastric pathology resembling human disease [166,167]. Alternatively, human-specific pathogens can be replaced by closely related mouse-pathogenic species. Since neither enteropathogenic nor enterohaemorrhagic *Escherichia coli* are able to naturally infect mice, *Citrobacter rodentium,* the only known pathogen of this attaching and effacing (A/E) family that naturally infects mice, is used to model the respective human diseases [168,169]. 

Although mice are no natural host for human clinical *S. aureus* isolates, our research group has shown that not only laboratory mice are naturally and persistently colonized with *S. aureus* but also wild mice and closely related species. Laboratory mice carry a large variety of *S. aureus* CCs, most of them likely originating from the human population. While CC88 spread across several continents for three decades, other CCs are sporadic introductions with limited expansion. The *S. aureus* populations in wild mice and related species are genetically more homogeneous than in laboratory mice but comprise unique lineages that are rarely isolated from other hosts [29,30,170,171]. 

Most laboratory and wild murine *S. aureus* strains show clear signs of adaptation to their host. They lack human-specific virulence factors, such as superantigens and the IEC, and barely possess any antibiotic resistance genes. Of note, a single CC130 isolate derived from a wild yellow-necked mouse carried the *mecC* gene, which underlines the role of wild mice as potential vectors for MRSA and argues for a One Health approach to combat the spread of MRSA. Moreover, some isolates show strong pro-coagulatory activity on murine plasma, as well as enhanced survival in murine blood, when compared to the human strain *S. aureus* Newman [97]. Notably, host-specific coagulation of plasma is also regarded as a host adaptive feature [58]. Active transmission of *S. aureus* between members of the same species is hard to investigate in wild mice; however, this has been shown for laboratory mice and their colonizing strains, so that actual host adaptation seems more likely than accidental spillover [30].

Table 2 gives an overview of potential host-adapted *S. aureus* strains that have already been described in the literature. Some of these will be discussed in detail in the following sections. The first mouse-adapted strain to be characterized in detail was JSNZ, a CC88 MSSA strain isolated in New Zealand from C57BL/6 mice suffering from preputial gland abscesses [171]. Female CD1 mice were more susceptible to experimental nasal and gastrointestinal colonization by *S. aureus* JSNZ than Newman. Furthermore, unlike *S. aureus* Newman, *S. aureus* JSNZ colonized mice without antibiotic pre-treatment, showing that it can compete with the natural microbiota of CD1 mice. Notably, this strain enables persistent intranasal colonization of mice, allowing researchers to study bacterial pathogenesis, novel decolonization drugs, or vaccines in a clinically more relevant mouse model. Sun et al. also isolated another mouse-adapted CC88 strain called *S. aureus* WU1 from male C57BL/6 mice suffering from preputial gland infections. Similar to JSNZ, this strain persistently colonized the murine nasopharynx and coagulated murine plasma better than human plasma [172]. Moreover, a CC15 isolate (SaF_1) originating from a colony of BALB/c mice also persistently colonized the gastrointestinal tract of BALB/c mice after environmental exposure. Interestingly, the authors observed the development of persistent and intermittent gastrointestinal carriage states in genetically identical mice, which points to the microbiome as a decisive factor in this setting [173]. 

While JSNZ is a superior colonizer, it shows a moderate virulence in infection models. In a bacteremia and pneumonia model in BALB/c mice, *S. aureus* JSNZ showed results comparable to *S. aureus* Newman in terms of virulence [97]. In contrast, in a renal abscess model, *S. aureus* JSNZ caused a more severe disease than *S. aureus* Newman [171]. In a subcutaneous infection model, *S. aureus* JSNZ showed a stronger upregulation of *saeR* transcripts (in relation to in vitro cultures) than *S. aureus* Newman [174], which might hint at a different virulence factor regulation in the mouse.

To test whether other mouse- and rodent-derived *S. aureus* isolates are more virulent in murine infection models, our group used a step-wise approach. After selecting prototypes for each CC, we selected candidates based on their pro-coagulatory activity and their ability to survive in murine blood. Subsequently, four mouse-adapted strains were compared to Newman in murine bacteremia and pneumonia models using BALB/c mice. Strikingly, a bank vole-derived CC49 isolate, named DIP, displayed a strongly increased virulence compared to *S. aureus* Newman in both models. For the bacteremia model, just one tenth of the inoculation dose of *S. aureus* Newman was sufficient to achieve comparable results with DIP [97]. This makes DIP an ideal candidate for murine infection models.

*S. aureus* LS-1 has been isolated from a swollen joint of a spontaneously arthritic NZB/W mouse [175,176] and has recently been characterized in detail [177,178]. It is often used in osteomyelitis models, but potential host adaptation has not yet been investigated [178,179]. *S. aureus* DAK is a murine mastitis isolate that was used for murine colonization models; however, no studies on its host adaptation have been performed yet [180]. 

Instead of using naturally adapted strains for the respective animal host, it is also possible to generate host-adapted strains in the laboratory by serial passaging in the desired host, an approach commonly used for viral pathogens [181,182,183,184]. Although bacterial evolution is much slower than viral evolution due to a lower mutation rate [185], it still seems feasible for experiments with bacteria as has been shown by Søndberg et al., who observed the adaptive evolution of *Salmonella enterica* serovar Typhimurium in infected mice within 2 to 4 weeks [186]. Similarly, McAdam et al. monitored the adaptive evolution of *S. aureus* in a cystic fibrosis patient over a period of 26 months and observed changes in phage content, as well as genetic polymorphisms in genes, which influence antibiotic resistance and global virulence regulation [187]. Moreover, the existence of hypermutable variants with an increased mutation rate under stressful conditions has been described for *S. aureus* [187,188,189]. Since a host switch puts *S. aureus* in such a condition, those variants might be candidates for fast adapting lineages.

The advantage of using host-adapted strains lies in their optimized interaction with the host compared to non-adapted strains, whose interaction is likely to be impaired by factors like receptor incompatibilities, superfluous or ineffective virulence factors, and the fact that the strain did not have time to co-evolve with the host’s microbiome and immune system. In the case of *S. aureus,* this enables adapted strains to persistently colonize mice, something that has been impossible to achieve so far [30], and allows researchers to lower infection doses up to ten-fold. Consequently, previously established models could be improved, e.g., decolonization models for testing new anti-infective drugs. Moreover, new experiments become possible, for instance those necessary for investigating the carrier–non-carrier dichotomy of *S. aureus*.

Nevertheless, there are also major drawbacks of using host-adapted strains. Most animal experiments are performed with the intention of ultimately transferring their results to the human system. Adapted strains and their interaction with the host might be less suitable to ensure sufficient transferability, since for instance, important virulence factors might be absent or inactive in the used pathogens. Moreover, the tissue anatomy, and thus, the niche ecology of the host might be completely different from that of humans. This problem is less relevant when investigating processes where homologues of interaction are known in the adapted system, such as LukMF’ as a bovine homologue of human-specific leukocidins targeting neutrophils [198], but becomes tedious if this is not the case, as in the case of human superantigens and the IEC in mice. Hence, instead of attempting to murinize *S. aureus*, it might be more sensible to humanize the mouse itself.

#### 4.2.2. Improving *S. aureus* Mouse Models by Using Humanized Mice

Humanized mice are often immunodeficient mice that express human gene products or are xenotransplanted with human cells, tissues, or organs [199]. Some of the first humanized mice were CB17 mice suffering from severe combined immunodeficiency due to a Prkdc^scid^ mutation, which received human peripheral blood lymphocytes [200]. Though humanized mice date back to the late 1980s, their use has only recently become more prevalent, since the first generations had to cope with serious limitations, including the reconstitution of just a limited set of human immune cells after xenotransplantation, as well as a very short life-span of the mice (<3 months) [201]. With respect to the humanization of the immune system in so-called human immune system (HIS) mice, continuous optimization has taken place in recent years. The humanization of HIS mice improved steadily from mice with “simple” genetic defects that led to immunodeficiency and to which human cells were xenotransplanted, to mice with more complex genetic modifications that received fetal liver, bone marrow and thymus tissue in addition to human stem cells, to additional knockins of human growth factors that supported the reconstitution of more immune cell populations. The history of humanized mice and their optimization [201,202,203] as well as an overview of available humanized mouse models, their general advantages and disadvantages, and their applications were excellently reviewed by others [199,204,205,206].

Although these fanciful tools are available nowadays, the use of humanized mouse models in infection models of *S. aureus* is still in its infancy [61]. Most published studies reported on humanizing the cellular receptors for human-specific staphylococcal toxins (Table 3), to enable cause-and-effect studies in the mouse model. Tromp et al. used mice whose complement C5a receptor 1 was humanized (hC5aR1) to demonstrate their increased susceptibility to *S. aureus* infection, as reflected by 10–100-fold higher bacterial loads in the skin, the kidneys, and the spleen compared to wildtype mice. This effect was mediated by the pore-forming toxin HlgCB, whose cellular receptor is C5aR1. Surprisingly, Panton–Valentine leukocidin (PVL), whose S subunit uses the same receptor, showed reduced activity, which led to the discovery that PVL’s F subunit requires specific interaction with human CD45 to work properly [207]. Species specificity of the virulence factor LukAB was shown by Boguslawski et al., who humanized LukAB’s receptor CD11b, resulting in enhanced susceptibility of humanized mice towards *S. aureus* bloodstream infection, as evidenced by a higher bacterial burden in the liver [145]. 

Another group xenotransplanted human skin onto SCID.bg mice and inoculated it with the MRSA strain USA300 SF8300. By measuring significantly upregulated human IL8 expression in skin biopsies, they were able to show the reaction of the human skin towards *S. aureus.* Moreover, they demonstrated that human IL8 is able to recruit murine neutrophils, which contribute to controlling MRSA growth on the human skin [208]. In another study, van Dalen et al. investigated the interaction of human langerin with *S. aureus* WTA. Using mice whose Langerhans cells constitutively express human langerin, they could show that following epicutaneous infection, high transcript levels of *Cxcl1*, *Il6*, and *Il17* were dependent on β-GlcNAc modifications on *S. aureus* WTA. Nevertheless, there was no difference in the bacterial burden of the lesions between wildtype and humanized mice, which might be attributed to the experimental design [152].

Another interesting approach to increase the susceptibility to *S. aureus* infection is to reconstitute mice with a humanized immune system [212,213,215]. For example, Prince et al. worked with nonobese diabetic (NOD)-*scid*-*IL2Rγ^null^* (NSG) mice lacking B, T, and NK cells and reconstituted them with a human hematopoietic system through fetal hematopoietic stem cell (CD34+) and thymic tissue grafts. They observed an increased severity of lung infection in these humanized mice, which could be attributed to the action of PVL [215] (Table 3). 

However, why are humanized mice, especially mice with a humanized immune system, not used more often in *S. aureus* research? The high costs of humanized mice and both the complexity and the time required for their generation can be limiting factors [71]. Moreover, there are serious limitations of humanized mouse models depending on the mouse strain used. Due to the lack of human growth factors, such as human IL3 or human GM-CSF, in most HIS mouse models, and the limited cross-reactivity of respective murine factors, the reconstitution of the innate immune cells remains incomplete and, hence, does not reflect the composition in humans. Although the situation can be improved by either injecting these factors directly or creating strains that produce them heterologously, such as MITRG and MISTRG strains, the impact on other parts of the immune system is still unknown, and adverse effects on the murine host, such as anemia, have been observed [205]. Since innate immune cells are crucial for the recognition of and defense against *S. aureus* and are therefore heavily manipulated by *S. aureus,* this significantly hampers *S. aureus* research [71,91,217]. If models utilizing stem cells for the reconstitution of the human immune system are employed without the co-transplantation of fetal human thymus tissue or without humanized MHC genes, T cells will be trained with murine MHC, leading to an unphysiological interaction with human APCs, and resulting in impaired class switching and disorganized secondary lymphoid structures [201]. Other factors that impair the efficient interaction of APCs with T cells are missing or poorly developed secondary lymphoid tissues, due to the absence of lymphoid tissue inducer cells in *Il2rg^–/–^* humanized mice [203,205] and different kinetics of the B and T cell reconstitution that require a precise timing of experiments [218].

Problems also arise from the partial humanization of mice. While some components of the mouse, such as the immune system, might be human, other parts are still murine. As a consequence, the virulence of *S. aureus* can be altered. For instance, the engraftment rate of human erythrocytes is generally quite low, while murine erythrocytes are actively destroyed by human innate immune cells, resulting in anemia of the mouse [201]. Since *S. aureus* can neither acquire sufficient amounts of iron from human erythrocytes (bad engraftment) nor from murine ones (host tropism of IsdB), its iron metabolism and therefore virulence will be heavily influenced [107]. On the host side, the interaction of the humanized and murine parts might be impaired. For example, homing of immune cells requires specific interactions of receptors and ligands both on immune cells and on tissue cells such as endothelial cells. If this interaction is disturbed by species-specificity, homing of immune cells will be compromised. This hampered interspecies cross-talk within a single organism could be the major disadvantage of humanized mouse models, because an efficient systemic immune response involves the interaction of many tissues and cell types via humoral and cellular factors, which cannot be guaranteed in such a system. Typical signs of this incompatibility are the frequently observed graft-versus-host disease in such animals, as well as their limited lifespan [204].

Despite these limitations, humanized mice still offer great opportunities for *S. aureus* research. First of all, the contribution of human-specific virulence and immune evasion factors such as superantigens and the IEC proteins to the virulence of *S. aureus* can be investigated *in vivo*. In addition, and similar to what has been previously published, the humanization of different parts of the mouse will provide information about the species specificity of *S. aureus* molecules and thus reveal factors essential for host adaptation, define their role in host tropism, and disclose their contribution to *S. aureus’* fitness in a certain host. Inspired by publications like those of Klicznik et al. and Dusséaux et al. in which not only the immune system but also other tissues, such as the skin or the liver, were humanized, it is also conceivable to use such a concept to study more localized *S. aureus* infections, such as skin infections, or *S. aureus*-associated diseases such as atopic dermatitis [219,220].

It has yet to be proven that a higher degree of humanization correlates with a better transferability of results to the human system. However, if this holds true, humanized mice might be a promising tool to address previously unsolved issues, such as the development of a successful *S. aureus* vaccine.

#### 4.2.3. Dirty Mice, Natural Microbiota, and Wildlings

The microbiome of an organism is an integral part of its physiology [221]. It plays a major role in shaping the host’s immune system during its development as well as later in life after it has matured [130]. Therefore, the composition of an organism’s microbiome not only affects the microbiota–host homeostasis and related health aspects, such as the susceptibility to autoimmune or allergic diseases [222,223], but also the host’s interaction with pathogens [224,225]. Laboratory mice are generally housed in a controlled and clean environment that is free of certain microorganisms. Thus, the microbiome of laboratory mice is very artificial, differing from that of wild mice or even humans [72,226]. Moreover the immune system of laboratory mice is in a naïve state as compared to the highly active and mature immune system of wild mice and adult human beings [227,228]. To date, *S. aureus* in vivo models have not taken the substantial impact of the microbiome into consideration. Recently, several approaches have been described to solve this problem: dirty mice, natural microbiota, and wildlings. 

In the dirty mouse approach, conventional laboratory mice are exposed to microorganisms that usually do not appear in breeding facilities. This exposure can either be very defined by infecting mice directly with known pathogens [229] or undefined by either co-housing them with “dirty” non-laboratory mice, such as pet-store mice [230], or exposing them to a natural environment [231,232]. The latter type of mouse is sometimes referred to as “rewilded”. Generally these procedures lead to significant changes in the mouse’s microbiome. Leung et al. observed an increase in Enterobacteriaceae, Lachnospiraceae, and Ruminococcaceae families in mice that were kept outdoors already after two weeks, while Clostridiaceae and Erysipelotrichaceae families were reduced. Moreover, their guts were also colonized by different fungal communities than those of laboratory mice [231]. Compared to laboratory mice, dirty mice also present a more mature phenotype of the immune system, as shown for instance by a higher amount of differentiated effector and memory T cells, more serum antibodies and class-switched B cells, a more “adult human being-like” transcription profile, and an increase in mucosally distributed T cells [230,231,232].

These changes culminate in a modified reaction of the immune system to antigenic stimuli. Reese et al. observed a reduced antibody response after vaccination against Yellow Fever Virus [229]; Beura et al. described an increased resistance to infection with *Listeria monocytogenes* and *Plasmodium berghei* ANKA. Furthermore, the differentiation of CD8+ T cells after Lymphocytic choriomeningitis virus infection was shifted from memory precursor to short-lived effector cells when compared to laboratory mice [230]; Leung et al. observed an increased gut nematode susceptibility in rewilded mice, which might be the result of a decreased type 2 and increased type 1 immune response in these mice [231]; Yeung et al. observed an increased cytokine production of mesenteric lymph node cells after ex vivo stimulation with various bacterial stimuli [232]. These examples clearly demonstrate that the exposure of laboratory mice to environmental microorganisms drastically changes their immunological phenotype, which in turn influences future experiments. Whether this is due to phenomena such as trained immunity, innate, and adaptive tolerance, antigenic mimicry, cross-reactivity of antigens, exhaustion of immune cells, or others remains to be clarified and will probably vary depending on the microorganisms used for exposure and then for the subsequent experiments. 

Looking at *S. aureus,* our research group has discovered that laboratory mice are frequently colonized with, and therefore pre-exposed to, *S. aureus* and hence mount a systemic immune response against it [30]. Most of the rewilded mice in Yeung et al.’s study were also positive for seemingly animal-adapted *S. aureus*, since they were lacking IEC-encoding prophages, but the authors could not exclude that *S. aureus* was already present in the laboratory mice before [232]. This has serious implications for *S. aureus* research, because such animals are no longer naïve towards *S. aureus*, and an uneven distribution of pre-exposed animals between control and experimental groups may largely distort results. Given the appropriate surrounding conditions, an endogenous *S. aureus* strain can also cause infections that can interfere with research findings, as recently observed in a rat infection model. In detail, in a model of delayed fracture healing using *S. aureus* ATCC49230 as infecting agent, *S. aureus* was also detected in 43% of the non-infected fracture group, with two rats even showing clear signs of bone infection. All *S. aureus* isolates from non-infected rats belonged to the same *spa* type (t1754), which differed from that of the infecting strain (t021), suggesting that this strain persisted in the rat colony and was introduced into the bone either during surgery, or through the animals while fiddling with their own surgical wounds [233]. Nevertheless, pre-exposure of experimental animals might be the right way forward for *S. aureus* research and potential vaccine development, as the majority, if not all, human beings are exposed to *S. aureus* early in life and hence immunologically primed [113,234,235,236,237]. 

A major disadvantage of dirty mice is the fact that exposure to environmental microorganisms occurs late after birth, so that they cannot participate early in shaping the physiology and immune system of the mouse. As a consequence, co-adaptation of microbiota and mouse might be impaired, and the created phenotype of the mouse might be more “infected” than “adapted”. 

This problem is dealt with using the natural microbiota approach in which the microbiome of selected wild mice is transferred to germ-free, pregnant mice, so that the offspring is already influenced in utero and naturally colonized after birth [72]. The transferred microbiome differs from that of laboratory mice and is stable over several generations. Mice generated in this way showed an increased survival to otherwise lethal influenza A virus infection, with reduced viral titers in the lung and less immune-mediated lung damage compared to conventional, barrier-raised C57BL/6 mice, as well as mice that originated from germ-free pregnant mice that received a microbiome from specific-pathogen-free C57BL/6 mice. Cytokine measurements in lung tissue revealed a shift towards anti-inflammatory cytokines, thus avoiding excessive inflammation with subsequent death. Moreover, natural microbiota mice exhibited improved resistance to mutagen/inflammation- induced colorectal tumorigenesis, which was shown by reduced weight loss after induction, a lower number, and less invasive tumors [72]. As with dirty mice, the parent generation of natural microbiota mice did not co-evolve with the introduced microbiome, which might have physiological consequences for the subsequent generations that have not been investigated so far. Additionally, the microbiome of other barrier sites such as the nose, skin, lungs, or the urogenital tract was neglected, so that this approach still remains incomplete. 

To optimize the natural microbiota model, the wildling approach was recently developed by Rosshart et al. [238]. In this model, C57BL/6 embryos were transferred into pseudopregnant wild mice and later on used for further breeding. This ensured that relevant microbiota and pathogens are considered at all mucosal barrier sites, that the physiology of the parent generation is adapted to them, and that the offspring is affected during all stages of embryonic and fetal development, as well as immediately after birth and throughout life, by natural colonization. Unlike the laboratory microbiome, the wildling microbiome was stable over several generations, and was able to recover after antibiotic treatment. In co-housing experiments of mice with a wild microbiome and a laboratory microbiome, respectively, the former outcompeted the latter and established itself in the laboratory microbiome group, which showed its superior adaptation to the murine host. Considering the immune system, the principal component analysis of the expression of immune-related genes in blood mononuclear cells put wildling mice closer to wild mice than laboratory mice. The same was true for the immunophenotype in the spleen, whereas the immunophenotypes of the skin and vagina were similar to those of laboratory mice. This might reflect the differential influence of the host’s microbiome and genome on different sites of the body. Indeed, one has to keep in mind that wildlings still possess the genome of laboratory mice, while they have a microbiome very similar to that of wild mice. 

Having established and characterized the wildling model, Rosshart et al. reproduced two human clinical studies to test whether wildlings are superior to conventional mice in predicting the outcomes of such studies. A CD28-superagonist, which activated and expanded anti-inflammatory regulatory T cells in conventional laboratory mice and showed promising results in autoimmune disease, inflammatory disease, and transplantation models, led to a life-threatening cytokine storm in human phase I clinical trials [239]. Unlike conventional laboratory mice, wildlings showed no increase in regulatory T cells but significantly elevated serum levels of the pro-inflammatory cytokines IL1β, IL2, IL4, IL6, IFNγ, and TNFα, mirroring the results of the human trial. In addition, in a mouse model of lethal endotoxemia, TNFα blockade protected mice from death but failed to do so in septic patients, where even an increased mortality was observed [240]. Using wildlings, similar results as in human trials could be observed [238]. Thus, it seems that wildlings are better models for humans than conventional laboratory mice, which might hopefully provide significant insights into many human diseases in future research.

## 5. Conclusions

*S. aureus* is a pathobiont of humans as well as a variety of animals worldwide. Its clonal population structure separates this species in more or less related lineages with distinct properties, such as virulence, host spectrum, and host tropism. The continuously high prevalence of multi-resistant strains (MRSA/VRSA) in hospitals, along with the spread of highly virulent and resistant CA-MRSA within the community, and the zoonotic potential of *S. aureus* increase the need for effective anti-infective measures and new treatment strategies. Mice as the primary animal model for *S. aureus* research contributed significantly to advances in understanding the pathogenesis of *S. aureus*, its interaction with the immune system and the development of potential treatment strategies but also failed as tools to produce an urgently needed vaccine. Promising results in mouse models have so far always proven to be inefficient in humans, which clearly shows the lack of transferability between some mouse models and the human system. Differences in the microbiome, physiology, and immune system of mice and humans, the host tropism of *S. aureus* strains and related species-specific virulence factors are only some of the underlying reasons, which must be considered in the development of future, more reliable mouse models. 

Mouse-adapted *S. aureus* strains, which take into consideration the host tropism of *S. aureus*, humanized mice, which take into account the species differences of mice and humans, and microbiome-optimized laboratory mice, which consider the influence of the microbiota on the host’s physiology, are just a few examples for already established optimizations of mouse models that have proven to be superior to conventional mouse models in several experimental settings. Although all of these models are recognized as a step in the right direction, there is still plenty of room for improvement, since each of them only addresses single aspects that limit transferability between mice and humans. Combining and re-interpreting some aspects might be the way to generate mouse lineages with superior transferability properties. Recently, Lundberg et al. transplanted human microbiota in conventional C57BL/6 mice and observed difficulties in establishment and just weak stimulation of the murine immune system [241]. The introduction of a human microbiome in humanized mice might improve interaction of microbiota with the host and open up new horizons of research. Furthermore, the use of wildlings with mouse-adapted strains of *S. aureus* is also conceivable, although the question arises whether the transferability might be reduced rather than improved. Finally, it can be concluded that rigorous modifications of conventional mouse models and model *S. aureus* strains may be required to overcome obstacles in *S. aureus* research and to achieve critical goals, such as the development of an effective anti-staphylococcal vaccine.

## Figures and Tables

**Table 1 ijms-21-07061-t001:** The choice of the *S. aureus* strain impacts on disease severity and host response.

*S. aureus* Strains	Animal Model	Infection Model	Main Findings	Ref.
Newman, N315, COL, JKD6159, NRS384, 512, LAC, Cowan	C57BL/6 mice	colonization/infection (i.n.)	- Differences in provoked weight loss, nasal bacterial load and persistence of colonization - Newman: short-term colonization; JKD6159: long term colonization - JKD6159: reduced nasal inflammation, less neutrophil egress into the airways, and reduced neutrophil–bacteria association compared to Newman	[86]
8325-4, Staph 38	C57BL/6, BALB/c mice	keratitis	- Staph 38: higher virulence (higher slit-lamp examination scores, higher bacterial burden in the eyes, and a higher recruitment of neutrophils)	[104]
muCC8c, muCC88d, DIP, JSNZ, Newman	BALB/c mice	pneumonia (i.n.) and bacteremia (i.p.)	- Survival of mice in both models is strain-dependent - DIP: highest; Newman: intermediate virulence - DIP infection dose can be reduced 10-fold	[97]
478, 586, 1611a, 1679a, ATCC29213	BALB/c mice	sepsis	- Differences in the lethal dose	[98]
ATCC 25923, CA-MRSA strain	BALB/c mice	intradermal infection	- CA-MRSA strain: higher recruitment of cells to the infection site and the draining lymph nodes - ATCC 25923: higher production of pro-inflammatory cytokines in the draining lymph nodes	[103]
ST121, ST96	rabbit (*Oryctolagus cuniculus*)	intramammary infection	- ST121: more severe mastitis as shown by a higher bacterial counts, higher cell recruitment, and larger abscesses	[99]
8325-4, Newman, UMCR1, MW2, five clinical ocular isolates: 91-717, 3161-06, 30103, 177, T1	New Zealand White rabbits	eye infection	- UMCR1 is the only *S. aureus* strain that grows within the anterior chamber after topic application to the rabbit eye	[102]
diverse strains causing endovascular complications and belonging to CC5, CC8, CC15, CC30 or CC45	honeycomb moth (*Galleria mellonella*)	survival model	- Survival of larvae is strain-dependent - MRSA strains cause a higher mortality than MSSA strains	[100]
6850, USA300, LS1, SH1000, Cowan1	human epithelial and endothelial cells, keratinocytes, fibroblasts, osteoblasts	in vitro infection	- Strain-dependent variations of host cell infection rate, cytokine production and cell death	[101]

i.n.: intranasal; i.p.: intraperitoneal; CC: clonal complex; MRSA: methicillin-resistant *Staphylococcus aureus*; MSSA: methicillin-sensitive *Staphylococcus aureus*.

**Table 2 ijms-21-07061-t002:** Overview of experimentally used, host-adapted *S. aureus* strains.

Strain *	Original Host	Adaptive Features **	Experimental Findings	Refs.
JSNZ (CC88-MSSA)	laboratory mice (C57BL/6)	- Lack of superantigen genes, *pvl* genes, antibiotic resistance genes, and *hlb*-integrating Sa3int prophage (IEC) - Agglutinates mouse plasma more readily than human plasma	- Better colonizer of mice and more virulent in an intraperitoneal infection model than the human-derived strain Newman - No requirement of antibiotic pre-treatment to induce persistent colonization - Clinically relevant model to test vaccines and *S. aureus* decolonization drugs	[30,97,136,171,172,174,190]
WU1 (CC88-MSSA)	laboratory mice (C57BL/6)	- Lack of superantigen genes and *hlb*-integrating Sa3int prophage (IEC) - Allelic variant of *vwb* that appears to promote enhanced agglutination of mouse plasma	- Persistent colonization of the nasopharynx in mice - Colonization triggers serum IgG response in mice	[172]
DIP (CC49-MSSA)	bank vole (*Myodes glareolus*)	- Lack of superantigen genes, *pvl* genes, antibiotic resistance genes, and *hlb*-integrating Sa3int prophage (IEC)	- Increased virulence compared to *S. aureus* Newman in BALB/c bacteremia and pneumonia models - Significant reduction of the inoculation dose	[97]
SaF_1 (CC15-MSSA)	laboratory mice (BALB/c)	- Lack of superantigen genes, *pvl* genes, and IEC genes	- Short-term exposure to SaF_1 can result in persistent gastrointestinal colonization, but only in a fraction of animals - Vertical and horizontal transfer of SaF_1 between mice	[173]
DAK	laboratory mice	ND	- Colonization efficiency is similar to *S. aureus* Reynolds and Newman	[180]
LS-1	laboratory mice (NZB/W)	ND	- Induces rapid joint destruction with visible synovial hypertrophy within 24 h	[176,177,178,179]
ST121 (CC121)	rabbit	- Lack of IEC genes - Species-specific *dltB* mutation	-Causes skin abscesses in rabbits at very low inoculum levels	[99,191]
KH 171	rabbit	ND	- Colonizes rabbit epithelia and spreads to different body sites - Development of large abscesses with acute onset and slow involution	[192,193,194]
UMCR1	rabbit	ND	- Reproducibly induces conjunctivitis after injection into the intact anterior chamber without the use of spermidin - Development of extensive disease and tissue damage in the eyes and hemorrhaging of the iris - Resistance to the ocular host defenses	[102,195,196]
PIL69, PIL74, PIL77, B40	pig	ND	- Natural colonization of newborn piglets following artificial colonization of the sow’s vagina - Colonization is stable for at least 28 days	[197]

* if known, information on clonal complex and methicillin resistance status was added; ** *dltB:* D-alanyl-lipoteichoic acid biosynthesis protein B; ND: not determined; *pvl*: Panton–Valentine leukocidin; *hlb*: beta-hemolysin; IEC: immune evasion cluster; *vwb*: von Willebrand binding protein.

**Table 3 ijms-21-07061-t003:** Overview of humanized mouse models used in *S. aureus* research.

Mouse Strain	Modification	*S. aureus* Strain	Model	Main Findings Regarding Humanized Mice	Ref.
C57BL/6J	Hemizygous for human hemoglobin (hHb)	Newman	Bacteremia (i.v.)	- *S. aureus* binds hHb more efficiently than murine hemoglobin - Enhanced IsdB-mediated disease severity in hHb expressing mice - Increased susceptibility of hHb-expressing mice to systemic staphylococcal infection	[153]
C57BL/6	- Humanized MHCII: HLA-DR4-IE (DRB1*0401) - No endogenous murine MHCII	Newman	Bacteremia (i.v.)	- SEA-dependent Vβ skewing of T cells and enhanced bacterial loads in liver and heart - Increased production of pro-inflammatory cytokines in liver and blood - Increased SEA-dependent CD11b^+^ Ly6G^+^ neutrophil recruitment to the liver - Increased SEA-dependent formation of hepatic abscesses	[209]
C57BL/6	- Humanized MHCII: HLA-DR4-IE (DRB1*0401) - No endogenous murine MHCII	Newman, COL	Colonization (i.n.)	- Newman: deletion of *sea* leads to transiently higher nasal loads - COL: deletion of *seb* leads to higher nasal loads - *S. aureus* superantigens may be involved in regulating bacterial densities during nasal colonization	[210]
NOD-*scid*-*IL2Rγ**^null^* (NSG)	Human neonatal foreskin skin (1 cm^2^)	USA300 FPR3757 (LAC)	- Human skin infection (topical application)	- *S. aureus* infects the grafted human epidermis without major disruption of the epithelial barrier - Hyperkeratosis of the stratum corneum after infection - Infiltration of the skin by neutrophils - *S. aureus* infection induces autophagy in the human skin	[211]
NOD-*scid*-*IL2Rγ**^null^* (NSG)	- Human CD34+ umbilical cord blood cells or - Human polymorphonuclear leukocytes (hPMN)	CST5	- Skin and soft tissue infection (s.c.)	- Enhanced susceptibility to *S. aureus* skin and soft tissue infection - 10–100-fold lower infection dose required - PVL induces dermonecrosis in NSG mice adoptively transferred with hPMN - PMX53, a human C5aR inhibitor, reduces the size difference of lesions induced by the PVL^+^ and PVL^−^ *S. aureus* but also reduces recruitment of neutrophils and exacerbates the infection	[212]
NOD-*scid*-*IL2Rγ**^null^* (NSG)	Human CD34+ hematopoietic stem cells	PS80	Bacteremia (i.p.)	- More severe infection reflected by a reduced survival percentage, increased weight loss, and a more rapid increase in bacterial burden - Higher rate of T cell activation and apoptosis	[213]
FVB/N	- Humanized surfactant protein B (SP-B) C or T allele - Lack of murine SP-B gene	Xen36	Pneumonia (i.t.)	- Mice with human SP-B C allele are more susceptible to *S. aureus* pneumonia than mice with SP-B T allele, presenting increased mortality, lung injury, apoptosis and NF-κB expression	[214]
NOD-*scid*-*IL2Rγ**^null^* (NSG)	Human CD34+ stem cells isolated from fetal liver tissue	USA300 LAC	Pneumonia (i.n.)	- More severe infection shown by higher bacterial loads in airways and lungs - Knockin of human IL3 and human GM-CSF leads to improved myeloid cell reconstitution and the development of human alveolar macrophages in humanized mice and further increased the bacterial burden - Increased number of human immune cells correlates with increased severity of *S. aureus* infection - PVL targets the human macrophage population and thereby contributes to the pathogenesis of the infection	[215]
C57BL/6 N	Humanized C5aR1	- USA300 SF8300 - ST80	Bacteremia (i.p.)	- HlgCB-mediated increase in bacterial loads in spleen and kidney - Identification of human CD45 as a co-receptor for PVL	[207]
SCID/Beige	Human skin graft (1.5–2 cm^2^)	USA300 SF8300	- Transient human skin colonization (topical application)	- *S. aureus* transiently colonizes the outer stratum corneum of xenotransplanted healthy human skin - Colonization induces a local inflammatory response shown by production of human IL8 - Human IL8 can recruit murine neutrophils - Neutrophil depletion leads to a higher bacterial burden on the surface of the human skin	[208]
huLangerin-DTR mice	Human langerin on Langerhans cells	USA300	Epicutaneous infection	- Human langerin on murine Langerhans cells interacts with *S. aureus* via langerin-WTA interaction - High transcript levels of *Cxcl1*, *Il6*, and *Il17* after infection are dependent on β-GlcNAc modifications on *S. aureus* WTA	[152]
C57BL/6N	Humanized C5aR1	N.A.	Neutrophil recruitment assay	- Improved binding of CHIPS to humanized C5aR1 - Administration of CHIPS dampens C5a mediated neutrophil migration	[216]
C57BL/6J	Humanized CD11b	USA300 strain LAC	Bacteremia (i.v.)	- Improved binding of LukAB to humanized CD11b - Enhanced susceptibility to MRSA bloodstream infection shown by increased bacterial burden in the liver	[145]

N.A.: not applicable; i.v.: intravenous: i.n.: intranasal; s.c.: subcutaneous; i.p.: intraperitoneal; i.t.: intratracheal; C5aR1: complement C5a receptor 1; IsdB: iron-regulated surface determinant protein B; SEA: staphylococcal enterotoxin A; SEB: staphylococcal enterotoxin B; PVL: Panton–Valentine leukocidin; NF-κB: nuclear factor kappa B; GM-CSF: granulocyte-macrophage colony-stimulating factor; WTA: wall teichoic acid; β-GlcNAc: β-N-acetylglucosamine; CHIPS: chemotaxis inhibitory protein of *Staphylococcus aureus*.

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
