# Peer review of "Staphylococcus aureus Host Tropism and Its Implications for Murine Infection Models"

_ijms, 2020, doi:10.3390/ijms21197061_

Round 1

Reviewer 1 Report

The manuscript addresses the important topic of S. aureus tropism to a variety of hosts and the possibility of using animal models in research.The knowledge on this subject presented in the manuscript is clear and arranged in paragraphs and tables.

In my opinion, Table 2 is unnecessary.The subject of the publication concerns strains of S. aureus. Table 2 describes bacteria other than S. aureus and increases the volume of the already extensive publication. The table can be replaced with a few description sentences.

Author Response

Point 1: The manuscript addresses the important topic of S. aureus tropism to a variety of hosts and the possibility of using animal models in research. The knowledge on this subject presented in the manuscript is clear and arranged in paragraphs and tables.

Response 1: We thank the reviewer for the evaluation and the positive feedback.

Point 2: In my opinion, Table 2 is unnecessary. The subject of the publication concerns strains of S. aureus. Table 2 describes bacteria other than S. aureus and increases the volume of the already extensive publication. The table can be replaced with a few description sentences.

Response 2: We agree with the reviewer’s opinion on this point. We have therefore removed the table, and included only two representative examples (Helicobacter pylori and Citrobacter rodentium) in the text. (Lines 434-440)

Reviewer 2 Report

The manuscript titled “Staphylococcus aureus host tropism and its implications for murine infection models” is a well-written review that summarizes the challenges and limitations associated with the in vivo study of S. aureus using murine models. It’s an overall balanced and up to date review with recent developments in the field, and a good addition to the current literature. The manuscript is pleasant to read, and the tables are useful and clear.

I have some minor suggestions for improvement:

The manuscript describes the limitations of murine infection models and gives an updated summary of current literature supporting the already known fact that (conventional) mouse models are not suitable to study (human) S. aureus strains.  However, over the years, research groups have also been (un)successful in elucidating the role of different S. aureus virulence factors in other non-murine (rabbits, zebrafish, nonhuman primates etc) animal models. Even though the manuscript focusses on the murine model, it is worth briefly mentioning in the discussion (or under 4.2) that other animal models have been used before and are also (not) suitable to study S. aureus in vivo

Line 210. Under 4.1 Current problems with conventional S. aureus mouse models. Is there literature on the infectious dose of human S. aureus strains in (conventional) murine models compared to what is suspected to be the minimum infectious dose in humans? The relatively high inoculum needed to induce S. aureus mediated disease in mice might hide the importance of specific S. aureus mechanisms/virulence factors crucial in the initial stages of infection when CFU numbers are less. This could be a good addition to the paragraph.

Line 268. Perhaps remove “Speaking”

Line 309. The authors correctly state that neutrophils are the first line of defence against S. aureus. We know now that S. aureus secretes more than 35 immune evasion molecules, of which more than a dozen (possibly important) human-specific virulence factors, that directly or indirectly disrupt neutrophil function. These virulence factors specifically interact with human neutrophils and not with murine neutrophils. This can also be briefly addressed in 4.1.5 (see Koymans et al Curr. Top. Microbiol. Immunol 2016), emphasizing the authors' message of the limitations of conventional murine models when studying these human S. aureus strains.  

Line 316. The authors briefly mention a specific S. aureus toxin. However, there are many more that specifically interact with human receptors, in particular the leukocidins, that can’t be adequality studied in wt murine models. These leukocidins, an important and highly debated group of S. aureus virulence factors, can briefly be mentioned or refer too (for example Spaan et al Nat Rev Microbiology 2017, Tromp et al Frontiers in Microbiology 2020).

Line 388. Refrain from using unpublished data in the manuscript

Line 488. Besides showing increased loads in the kidney and spleen, the authors should also add “skin” to the sentence. Reason being that PVL has been associated with skin infections and therefore adds to the surprise that no PVL phenotype was observed during the experiments using hC5aRKI mice.

Line 619. The authors refer to their own study, however, that humans have been exposed to S. aureus and have circulating antibodies/humoral response against S. aureus virulence factors has been described in studies by Verkaik et al amongst others. Authors should also refer to these studies if possible.

Author Response

Point 1: The manuscript titled “Staphylococcus aureus host tropism and its implications for murine infection models” is a well-written review that summarizes the challenges and limitations associated with the in vivo study of S. aureus using murine models. It’s an overall balanced and up to date review with recent developments in the field, and a good addition to the current literature. The manuscript is pleasant to read, and the tables are useful and clear.

Response 1: We thank the reviewer for the positive and valuable feedback.

Point 2: The manuscript describes the limitations of murine infection models and gives an updated summary of current literature supporting the already known fact that (conventional) mouse models are not suitable to study (human) S. aureus strains.  However, over the years, research groups have also been (un)successful in elucidating the role of different S. aureus virulence factors in other non-murine (rabbits, zebrafish, nonhuman primates etc) animal models. Even though the manuscript focusses on the murine model, it is worth briefly mentioning in the discussion (or under 4.2) that other animal models have been used before and are also (not) suitable to study S. aureus in vivo.

Response 2: We agree that it is important to mention other animal models that have already been used in S. aureus research, and have added this information under 4.2. Since a thorough description of both advantages and limitations of all those animal models would be beyond the scope of this review, we believe that the added statements are sufficient to give a basic impression of this aspect of S. aureus research. (Line 411-430)

Point 3: Line 210. Under 4.1 Current problems with conventional S. aureus mouse models. Is there literature on the infectious dose of human S. aureus strains in (conventional) murine models compared to what is suspected to be the minimum infectious dose in humans? The relatively high inoculum needed to induce S. aureus mediated disease in mice might hide the importance of specific S. aureus mechanisms/virulence factors crucial in the initial stages of infection when CFU numbers are less. This could be a good addition to the paragraph.

Response 3: We thank the reviewer for pointing out this important aspect of infection models, and have included a new paragraph concerning this issue. As outlined in this paragraph the infectious dose has a major impact on the infection model and is therefore worth discussion. (Line 225 and 264-295)

Point 4: Line 268. Perhaps remove “Speaking”

Response 4: Done (Line 302)

Point 5: Line 309. The authors correctly state that neutrophils are the first line of defence against S. aureus. We know now that S. aureus secretes more than 35 immune evasion molecules, of which more than a dozen (possibly important) human-specific virulence factors, that directly or indirectly disrupt neutrophil function. These virulence factors specifically interact with human neutrophils and not with murine neutrophils. This can also be briefly addressed in 4.1.5 (see Koymans et al Curr. Top. Microbiol. Immunol 2016), emphasizing the authors' message of the limitations of conventional murine models when studying these human S. aureus strains. 

Response 5: Considering the importance of neutrophils/phagocytes for the immune response against S. aureus, we agree that it is worth to point this out more precisely. We have added two more examples and references to relevant literature.  (Line 361-365)

Point 6: Line 316. The authors briefly mention a specific S. aureus toxin. However, there are many more that specifically interact with human receptors, in particular the leukocidins, that can’t be adequality studied in wt murine models. These leukocidins, an important and highly debated group of S. aureus virulence factors, can briefly be mentioned or refer too (for example Spaan et al Nat Rev Microbiology 2017, Tromp et al Frontiers in Microbiology 2020).

Response 6: We added the information that in mouse models more leukocidins are either inactive or less active and referred to relevant literature. (Line 358-359)

Point 7: Line 388. Refrain from using unpublished data in the manuscript

Response 7: We have removed the information about unpublished data and have added information about already published data of our working group. (Line 452-454)

Point 8: Line 488. Besides showing increased loads in the kidney and spleen, the authors should also add “skin” to the sentence. Reason being that PVL has been associated with skin infections and therefore adds to the surprise that no PVL phenotype was observed during the experiments using hC5aRKI mice.

Response 8: For the reason mentioned by the reviewer, we have added the information that also the skin showed increased bacterial loads. (Line 553)

Point 9: Line 619. The authors refer to their own study, however, that humans have been exposed to S. aureus and have circulating antibodies/humoral response against S. aureus virulence factors has been described in studies by Verkaik et al amongst others. Authors should also refer to these studies if possible..

Response 9: We have added more relevant references. (Line 690)

Other changes:

Lines 225-227: updated enumeration

Line 297: updated enumeration

Line 333: updated enumeration

Line 350: updated enumeration

Line 390: updated enumeration

Lines 433-434: replaced “aforementioned” by “mentioned in 4.1.”

Line 459: Since the original Table 2 was removed “Table 3” was replaced by “Table 2”

Line 511: Since the original Table 2 was removed “Table 3” was replaced by “Table 2”

Line 550: Since the original Table 2 was removed “Table 4” was replaced by “Table 3”

Line 566: Since the original Table 2 was removed “Table 4” was replaced by “Table 3”

Line 580: Since the original Table 2 was removed “Table 4” was replaced by “Table 3”

Lines 680-687: To further support our statement that colonizing strains may have an influence on the experimental outcome, we added some information about a study by Helbig et al. in which 43% of the animals in the control group were unexpectedly infected with a suspected endogenous strain. Although rats were used in this study, we think that such things can easily happen in mouse models as well.

Line 781: Added CFU to the abbreviation table